# Are Football Players More Prone to Muscle Injury after COVID-19 Infection? The “Italian Injury Study” during the Serie a Championship

**DOI:** 10.3390/ijerph20065182

**Published:** 2023-03-15

**Authors:** Alessandro Corsini, Andrea Bisciotti, Raffaele Canonico, Andrea Causarano, Riccardo Del Vescovo, Pierluigi Gatto, Paolo Gola, Massimo Iera, Stefano Mazzoni, Paolo Minafra, Gianni Nanni, Giulio Pasta, Ivo Pulcini, Stefano Salvatori, Marco Scorcu, Luca Stefanini, Fabio Tenore, Stefano Palermi, Maurizio Casasco, Stefano Calza

**Affiliations:** 1Italian Sport Medicine Federation (FMSI), 7000196 Rome, Italy; 2Kinemove Rehabilitation Center, 54027 Pontremoli, Italy; 3Società Sportiva Calcistica Napoli, 81030 Napoli, Italy; 4Associazione Sportiva Roma, 00182 Rome, Italy; 5Hellas Verona Football Club, 37126 Verona, Italy; 6Genoa Cricket and Football Club, 16155 Genoa, Italy; 7Spezia Calcio, 19136 La Spezia, Italy; 8Football Club Crotone, 88900 Crotone, Italy; 9Sport Medicine Department—Milan Lab, Associazione Calcistica Milan, 20149 Milano, Italy; 10Torino Football Club, 10121 Torino, Italy; 11Bologna Football Club 1909, 40132 Bologna, Italy; 12Parma Calcio, 43123 Parma, Italy; 13SS Lazio, 00060 Rome, Italy; 14Benevento Calcio, 82100 Benevento, Italy; 15Cagliari Calcio, 09124 Cagliari, Italy; 16Juventus Football Club, 10151 Torino, Italy; 17Udinese Calcio, 33100 Udine, Italy; 18Public Health Department, University of Naples Federico II, 80131 Naples, Italy; stefanopalermi8@gmail.com; 19Unit of Biostatistics and Bioinformatics, Department of Molecular and Translational Medicine, University of Brescia, 25121 Brescia, Italy

**Keywords:** COVID-19, football, muscle injury, risk factor, epidemiology

## Abstract

Introduction: Football was the first sport to resume competitions after the coronavirus disease 2019 (COVID-19) lockdown and promptly the hypothesis was raised of a potential relationship between the severe acute respiratory syndrome coronavirus 2 (SARS-CoV-2) infection and musculoskeletal injuries in athletes. This study aimed to confirm the association between SARS-CoV-2 infection and muscle strain injury in a large population of elite football players and to investigate if the COVID-19 severity level could affect the risk of injury. Methods: A retrospective cohort study involving 15 Italian professional male football teams was performed during the Italian Serie A 2020–2021 season. Injuries and SARS-CoV-2 positivity data were collected by team doctors through an online database. Results: Of the 433 included players, we observed 173 SARS-CoV-2 infections and 332 indirect muscle strains. COVID-19 episodes mostly belonged to severity level I and II. The injury risk significantly increased after a COVID-19 event, by 36% (HR = 1.36, CI_95%_ 1.05; 1.77, *p*-value = 0.02). The injury burden demonstrated an 86% increase (ratio = 1.86, CI_95%_ 1.21; 2.86, *p*-value = 0.005) in the COVID-19 severity level II/III versus players without a previous SARS-CoV-2 infection, while level I (asymptomatic) patients showed a similar average burden (ratio = 0.92, CI_95%_ 0.54; 1.58, *p*-value = 0.77). A significantly higher proportion of muscle–tendon junction injuries (40.6% vs. 27.1%, difference = 13.5%, CI_95%_ 0.002%; 26.9%, *p*-value = 0.047) was found when comparing level II/III versus Non-COVID-19. Conclusions: This study confirms the correlation between SARS-CoV-2 infection and indirect muscle injuries and highlights how the severity of the infection would represent an additional risk factor.

## 1. Introduction

Since the beginning of 2020, the world has been experiencing the coronavirus disease 2019 (COVID-19) pandemic [1], caused by the severe acute respiratory syndrome coronavirus 2 (SARS-CoV-2), which led to suffering and deaths, changing almost every aspect of our society. Sports also had to kneel and stop all activities for a long time. The fear that sports could represent a risk factor in virus spread [2,3], disease aggravation [4], and long-term health consequences [5,6,7] has prompted sports doctors to proceed based on the principle of maximum prudence [8]. Football was the first sport to resume competitions after the lockdown started in March 2020. During the interruption, sport medicine physicians discussed issues related to training during the lockdown [9], tried to promote the maintenance of physical health, and mitigate the increased risk of injury expected upon resumption. Subsequently, the scientific literature debated the issue of return to play, trying to figure out which screening protocol could best assess the consequences of the infection, especially the cardiological, hematological, and pneumological ones [10,11,12].

In the meantime, football institutions, in collaboration with international medical societies, have proposed screening and organizational protocols to be implemented during national and international competitions [13]. The Fédération Internationale de Football Association (FIFA) changed the rules relating to player substitutions during matches, taking them from three to five for each team per match [14]. The sports season 2020/21 started at different times in European countries. Furthermore, some teams had a shorter rest time after the end of the previous season, due to the delayed end of the European Champions League and Europa League competitions. The Italian Championship started on 19 September 2020 and quickly had to deal with a resurgence of the pandemic, with a steep increase in positive players a few months after the start. All these factors significantly influenced the performance of the championships in all countries of the world.

Several studies tried to investigate how SARS-CoV-2 infection could affect the performance of athletes and, at the same time, put forward the hypothesis that COVID-19 could increase the risk of musculoskeletal injuries [5,9,10,15,16,17,18,19]. Indeed, muscle injuries still represent the first cause of injuries in football, accounting for a huge amount of missed matches for the players [20,21]. However, to the best of our knowledge, only a few studies have focused on the association between SARS-CoV-2 infection and muscle strain injuries in elite football players [22,23,24,25,26,27,28,29], often with questionable and conflicting results [30]. Furthermore, none of these studies investigated the influence of the COVID-19 level on the risk of injury.

Therefore, the purpose of our study was to confirm whether SARS-CoV-2 infection was associated with an increase in the number of muscle injuries in a large population of professional athletes, and to investigate if the COVID-19 severity level could affect the risk of injury.

## 2. Materials and Methods

### 2.1. Study Design

The Italian Injury Study is a retrospective observational cohort study promoted by the Italian Sports Medicine Federation (FMSI). At the end of the 2020/2021 football season, all teams belonging to the Italian Serie A (the highest professional football league in Italy) were invited to participate in the study, collecting the required data retrospectively.

Teams that agreed to participate in the Italian Injury Study were asked to individuate a member of the medical staff who would be responsible for collecting the data, named Team Data Manager (TDM). All the TDMs participated in a preliminary meeting in which the methodology of the study and the terms of data acquisition were described and discussed in detail. The collecting procedure was performed according to the international consensus statements for epidemiological study of injuries to professional football players [31,32]. TDMs of each football club compiled an online database provided with a standardized form that would gather all the data necessary for the study. All the information collected was fully anonymized: the teams, players and TDM identities were coded using computer-generated random hash strings. The platform was compliant with Italian laws relating to privacy and data protection. All the players involved were informed about the purpose of the study and were required to provide written informed consent. The study was conducted in accordance with the Declaration of Helsinki principles and its later amendments.

The 4 June 2021 has been identified as the last date for the delivery of the final data requested. The injury data collection period stretched from 22 August 2020 to 23 May 2021.

The data relating to positivity for COVID-19 were collected, starting on 1 January 2020. During that period, a SARS-CoV-2SARS-CoV-2 infection was tested using a PCR nasal swab performed and analyzed by an independent contractor. Each player was tested at least 48 h before each official game during the whole season, in case of symptoms or teammates’ positivity. Isolation of 14 days at home was mandatory. A negative test was necessary to return to training.

### 2.2. Patient and Public Involvement

We aimed to collect an exhaustive sample of Italian Serie A. Unfortunately, five teams did not agree to participate. The 15 medical staff who agreed to participate were involved in the injury registration and the SARS-CoV-2 infection data collection. This study was carried out without patient involvement. Athletes were not invited to actively contribute to the study design or to write or edit the present document. However, each TDM and all team players provided consent to analyze the COVID-19-related data and they were all informed of the study results.

### 2.3. Setting

The study group consisted of all the players belonging to the Italian Serie A who had agreed to participate in the study and who had regularly taken part in the sporting activity throughout the observation period.

### 2.4. Inclusion Criteria

The study included all the players for whom the following data were available:SARS-CoV-2 infection characteristics, defined as [33]:
Polymerase Chase Reaction (PCR) COVID-19 test positivity and subsequent PCR negativization date;Return to play (RTP) date;Clinical classification of COVID-19 severity.
2.Injuries:
Injury occurrence and RTP date;Anatomical location of the injury;Situation in which the injury occurred (i.e., training or competition);Type of injury (i.e., first injury or re-injury).
3.Exposure:
Day-by-day players’ exposure time in training or match.

### 2.5. Exclusion Criteria

Those players for whom all the aforelisted data were not available for the entire observation period were excluded from the study.

### 2.6. Subjects

Fifteen clubs out of twenty belonging to the Italian Serie A agreed to participate in the study.

Players who left or joined the team during that season were included during their time spent on the team.

### 2.7. Data Collection

The following data were recorded:The number of subjects who contracted COVID-19 during the observation period. For each subject affected by SARS-CoV-2 infection, the days lost due to infection (RTP time) and the severity of the disease were also collected. The severity was classified based on five levels according to the Coronavirus Disease 2019 (COVID-19) Treatment Guidelines [34] (Table 1). RTP was defined as the moment when a player made a full return to training and competition, without any restrictions [35,36].

2.The total number of indirect muscle injuries that occurred during training and competition. For these muscle strains, the team’s physician made the final diagnosis and this was supported with medical imaging (ultrasound or MRI) and correctly classified [20]. The time-loss injury, the injury anatomical location, and the site of injury within the muscle were also registered. Other types of acute injuries, overuse injuries and illness were not considered events of interest. Each indirect muscle injury was classified as COVID-19 or Non-COVID-19 depending on whether it was preceded by a positive SARS-CoV-2 test during the observational period.3.For each subject, the match or training exposure time (ET) [37].4.The injury burden (number of days lost to injury per 1000 h of player exposure injuries) for each injured subject.5.The number of days between RTP and the first recorded indirect muscle injury.

### 2.8. Statistical Analysis

Quantitative data were reported as mean values and standard deviation (SD), while categorical data were summarized using counts and percentages. Injury risk was modeled as a repeated event, while COVID-19 status was entered as a time-varying exposure, using a Prentice, Williams, and Peterson (PWP) model.

The injury incidence rate was modeled using a Generalized Linear Mixed Model (GLMM) [38] assuming a Poisson distribution for injury counts on the player and exposure time entered as an offset term.

Injury burden was defined as the total amount of days lost due to injuries for each player, scaled to 1000 h/player of exposure. Injury burden was estimated by modeling injury time for each patient using a GLMM with exposure time entered as an offset term. Due to the highly skewed distribution and zero inflation, we used a Tweedie distribution family [39] that allows us to seemingly model skewed data with a point mass at zero. The usage of GLMM models allows one to account for the potential within player event correlation, as players infected by SARS-CoV-2 contribute both to Non-COVID-19 (before infection) and COVID-19 groups.

The proportions of injuries according to the anatomical location were estimated using a multinomial regression model.

The comparison of average COVID-19 duration between gravity levels (II–III vs. I) was performed using a robust linear model, to account for some extreme values [40].

Results were reported as estimates and corresponding 95% confidence intervals, with *p*-value adjusted for multiple comparisons when needed. All statistical tests were two-sided and assumed a 5% significance level. All analyses were performed using R (version 4.2.1) [41].

## 3. Results

Fifteen out of twenty teams belonging to the Italian Serie A championship 2020/2021 participated in the study. According to the eligibility criteria, 433 subjects (mean age ± SD of 26.08 ± 5.03 years) were included in the study. During the follow-up period, we observed 173 subjects (39.95%) with SARS-CoV-2 infections, a total of 332 indirect muscular injuries affecting 204 players, and 214 other events such as sickness that was not COVID-19 related or injuries other than the one of interest.

Among the indirect muscular injuries, 104 (31.3%) occurred after a SARS-CoV-2 infection, while 228 (68.7%) were not preceded by COVID-19 episodes. COVID-19 episodes were mostly level I (78, 45.1%) and II (84, 48.6%), with a few level III (11, 6.4%). Due to the low number of moderate illness cases, we collapsed levels II and III as Mild-Moderate illnesses.

The risk of injury following a COVID-19 episode was estimated using a Cox model for repeated events (injuries) and time-varying exposure (COVID-19 disease). The risk of injury significantly increased after a COVID-19 event by 36% (HR = 1.36, CI_95%_ 1.05; 1.77, *p*-value = 0.02) (Table 2).

When considering the injury incidence, expressed as the number of injuries per 1000 h of exposure (playing time), we estimated a 69% increase in the injury incidence rate (RR = 1.69, CI_95%_ 1.21; 2.38, *p*-value = 0.0023) in the Mild-Moderate level versus Non-COVID-19, while level I COVID-19 (asymptomatic) showed a rate comparable to the Non-COVID-19 group (RR = 0.95, CI_95%_ 0.61; 1.46, *p*-value = 0.80) (Table 3).

The injury burden showed a similar pattern, with a substantial and significant increase in Mild to Moderate COVID-19 compared to Non-COVID-19, with an 86% increase (ratio = 1.86, CI_95%_ 1.21; 2.86, *p*-value = 0.005), while asymptomatic subjects showed a similar average burden (ratio = 0.92, CI_95%_ 0.54; 1.58, *p*-value = 0.77) (Table 4).

To evaluate if the proportion of injuries in the muscle–tendon junction might vary depending on the injury being after a SARS-CoV-2 infection or not, and on the level of the COVID-19 symptoms, we tested if the proportion of muscle–tendon junction events, estimated using a multinomial model, was significantly different according to COVID-19 status. We found a significant, although marginally, higher proportion of muscle–tendon junction injuries (40.6% vs. 27.1%, difference = 13.5%, CI_95%_ 0.002%; 26.9%, *p*-value = 0.047), when comparing level II/III versus Non-COVID-19. Again, level I COVID-19 was not significantly different from Non-COVID-19 (35.3% vs. 27.1%, difference = 8.2% CI_95%_ −8.9%; 25.3%, *p*-value = 0.35) (Table 5).

According to the Italian law during the study period, the minimum isolation period after infection was 14 days regardless of the severity of the symptoms; we computed that the average number of days of absence due to SARS-CoV-2 infection was equal to 19.0 (robust mean = 16.1) days (range 14–63), with no statistical difference among COVID-19 levels (robust t-test. Average difference II/III vs. I: −0.023, CI_95%_ −1.16; 1.11, *p* = 0.97) (Table 6).

## 4. Discussion

The results of this study show that the football players affected by SARS-CoV-2 infection had a 36% higher risk of having indirect muscle injuries than players who never contracted COVID-19 or whose indirect muscle injuries occurred before contracting the infection. Furthermore, the injury incidence increases in the COVID-19 severity level: despite the COVID-19-unrelated group having a similar muscular injury incidence to the players that suffered level I COVID-19, the players having II and III COVID-19 severity levels had a 69% incidence increase in indirect muscle injuries.

These data are confirmed by the injury burden parameter, the cross-product of severity and incidence, that recently was suggested to provide a more complete picture of risk assessment in sports injuries [42,43]. Indeed, the data of our study show an injury burden increase in the II and III COVID-19 severity levels compared to the COVID-19-unrelated group.

Despite COVID-19 primarily affecting the cardio-respiratory system [44], the SARS-CoV-2 infection may affect muscle function, compromising peripheral capillarization and consequently decreasing muscle oxygen uptake, limiting oxidative metabolic pathways [45,46,47]. This limitation in the aerobic metabolic system may, in an intermittent high-intensity sports activity such as football, prematurely induce a state of fatigue, which represents a predisposing factor for indirect muscle injury [48,49]. Indeed, fatigue can alter the muscle recruitment pattern and force production, thereby increasing muscle injury risk [48]. Furthermore, the reduction in the oxidative capacity of the muscle [50] increases the involvement of the anaerobic lactacid metabolism with the same effort intensity, giving rise to subsequent premature muscle acidosis [51]. An increase in muscle acidosis leads to the greater fragility of the muscle fibers and subsequently to a higher risk of muscle injury [52,53].

Moreover, aerobic system perturbation may reduce the overall endurance capacity of the athlete. Since a lower VO_2_ max is an independent risk factor for indirect muscle injuries [54], we may suppose that an athlete who has contracted COVID-19 is subject to an increased risk of muscle injury.

A further risk factor that could play a role in the increased injury incidence was the isolation period after the SARS-CoV-2 infection. As a matter of fact, during the 2020–2021 season, Italian law mandated the 14-day isolation to be spent at home. Indeed, the detraining induced by such an isolation period may represent another important risk factor for indirect muscle injuries [55,56]. This hypothesis is consistent with other studies showing that muscular detraining may induce some important physiological adaptations [57]. Some authors [58] demonstrated how, after a muscle activity suspension of 14 and 23 days, the knee extensor torque, the cross-sectional, the vastus lateralis fascicle length and the tendon stiffness decreased. Furthermore, a significant deterioration in tendon mechanical properties occurring within two weeks, exacerbating in the third week of suspension of muscle activity, was noted. The authors conclude that rehabilitation aimed to limit muscle and tendon deterioration should probably start within 2 weeks of unloading.

These effects on muscle structure could be the cause of a muscular force and structural imbalance that produced the shift in the incidence of injury site location within the muscle structure, despite the type of muscle not differing in the COVID-19 and Non-COVID-19 groups. Indeed, we found increased susceptibility to injury of the muscle–tendon junction in the COVID-19 level II and III than in the COVID-19-unrelated group. These findings are in line with some other studies that demonstrate how immobilization and detraining can reduce the MTJ structure and endurance in rats [59,60,61] and humans [62]

Yet, the data in our study do not allow us to assert that a longer RTP duration is related to the level of severity of SARS-CoV-2 infection. Probably the need to have the players back on the team as soon as possible leveled the RTP duration regardless of the COVID-19 illness severity. Furthermore, it is important to remember that after COVID-19, a period of rest until the complete resolution of symptoms [56,63], followed by a specific athletic reconditioning period [55], is strongly recommended. Unfortunately, due to team needs, this is often impossible in the middle of a professional sporting season [51,64]. Indeed, most of the players, once recovered from COVID-19, were generally reintegrated into the team as soon as possible, to respond to team performance needs. During the post-COVID-19 period, it is probable that the players not only did not go through a suitable physical reconditioning period but were forced to train and compete with players who, not having contracted COVID-19, had never stopped training. Consequently, they were not in suitable physiological conditions for being exposed to the same training volumes and intensity as the other players, which increased the risk of injury. This hypothesis is consistent with a study conducted on football players belonging to Bundesliga during the 2019–2020 season, showing a proportionally higher number of injuries in athletes striving to play their first match too early after the COVID-19 disease [65]. The effect of a quick RTP could explain the susceptibility of injury in matches compared to training recorded in the level II and III groups. We can argue that the higher intensity level and the strong muscle demand of matches could reveal the low level of training due to the COVID-19 illness.

The data recorded in our study show that the effect of the severity of COVID-19 level on the time elapsed between the RTP period and the first recorded indirect muscle injury was not statistically significant. Furthermore, the athletes who suffered a COVID-19-related injury (severity levels II and III) had a more severe injury burden in comparison to the athletes who suffered a COVID-19-unrelated injury. In addition to this, the injury burden value of athletes affected by COVID-19 progressively increased according to the three levels of infection severity recorded.

The higher value of injuries incidence and injury burden, regardless of the RTP time, for athletes affected by COVID-19 level II and III as compared to those affected by COVID-19 level I, may be at least partially argued by the musculoskeletal susceptibility to SARS-CoV-2 infection. This infection seems to influence muscular cell activity by a direct mechanism, with the virus binding to the ACE2 receptor on the skeletal muscle cell surface, and by an indirect mechanism named “cytokine storm”, a deregulated release of numerous cytokines by the immune system after a lung infection [47]. The process triggered by the virus could induce muscle fiber proteolysis, promoting a decrease in protein synthesis, interfering in the myogenic process, and disrupting the body’s homeostasis [66,67].

However, the lack of scientific research focusing on the musculoskeletal system of athletes, and the fact that few studies included functional or morphological techniques in their methodologies to demonstrate the direct muscle functional damage, prompt the authors to remain cautious in conferring a causal link.

Our results confirm the findings of other studies [23,51,65], even if our study has more exhaustive collecting modalities, a bigger sample size, and a higher number of events recorded. Other similar research explores this situation in other major football European championships, demonstrating how this is a new, hot, and widespread topic in the scientific literature: Wezenbeek et al. [26] reported a five times higher risk of developing a muscle strain after COVID-19 in Belgian professional footballers; Mannino et al. [22] showed an increase in injuries in the pandemic-related English Premier Leagues season; Maestro et al. [19] highlighted a doubled risk of muscle injuries after COVID-19 in Spanish football players; and Seshadiri et al. [25] shared a similar worrying scenario in German Bundesliga.

The clinical consequences of the situations described in our study would recommend extreme caution in the process of reintegration into training and competitive activity of a player who has contracted the SARS-CoV-2 infection [68], especially after moderate and severe ones. Aerobic, resistance, and speed training should follow specific phases, based on the progressiveness of the training load and the consequent physiological adaptation response [55]. Moreover, muscle injury prevention exercises should be introduced and/or increased as part of the training program’s response [55]. Finally, a gradual and cautious return to training and return to play protocol, under medical supervision, should be adopted [68].

This study suffers from some limitations. The most important is represented by the fact that our study design did not allow us to make direct causal inferences concerning the effects of COVID-19 pathology. In addition, the associations found in this study should be interpreted considering that many interdependent factors are involved in causing muscle injuries. The study does not consider, for example, that pharmacological therapies (such as steroids and antibiotics) taken for the treatment of COVID-19 may have played a role in making the player more susceptible to muscle injury. Moreover, no follow-up was performed to confirm this trend in the following seasons, with different variants of the virus, different pharmacological and diagnostic approaches to the pandemic, and new rules about isolation. Finally, we must clarify that injuries can be a repeating event within subjects, and the same subject might have one or more injuries and these can occur both before and after COVID-19. Therefore, a subject contributes to both groups, without and with COVID-19, depending on the time point considered. Thus, this set of players at risk of injuries is a dynamic riskset.

Therefore, the results of the study must be interpreted with caution.

## 5. Conclusions

Our results show the correlation between COVID-19 infection and an increase in indirect muscle injuries in professional football players. Furthermore, this study highlights how the severity of the infection would represent an additional risk factor. Moreover, we demonstrated how COVID-19 level I infection does not seem to affect the risk of muscle injury more than normal football activity. Considering that no difference in time to RTP after infections was found, these data suggest that the short-term detraining effects due to the time loss, but probably also a direct action of the virus and the inflammatory process triggered by the virus on muscle tissue, could be associated with a greater risk of indirect muscle lesions.

However, the continuous evolution of the virus and the lack of studies focused on musculoskeletal system damage is preventing us from drawing definitive conclusions. More studies are needed to clarify the role of COVID-19 in causing football muscle injuries.

## Figures and Tables

**Table 1 ijerph-20-05182-t001:** Clinical level of COVID-19 disease severity.

COVID-19 Severity Level		Type of Patients
I	Asymptomatic or Presymptomatic Infection:	Individuals who test positive for SARS-CoV-2 using a virologic test (i.e., a nucleic acid amplification test or an antigen test) but who have no symptoms that are consistent with COVID-19.
II	Mild Illness	Individuals who have any of the various signs and symptoms of COVID-19 (e.g., fever, cough, sore throat, malaise, headache, muscle pain, nausea, vomiting, diarrhea, loss of taste and smell) but who do not have shortness of breath, dyspnea, or abnormal chest imaging.
III	Moderate Illness	Individuals who show evidence of lower respiratory disease during clinical assessment or imaging and who have a saturation of oxygen (SpO2) ≥ 94% on room air at sea level.
IV	Severe Illness	Individuals who have a SpO2 < 94% on room air at sea level, a ratio of arterial partial pressure of oxygen to fraction of inspired oxygen (PaO2/FiO2) < 300 mmHg, respiratory frequency > 30 breaths per minute, or lung infiltrates > 50%.
V	Critical Illness	Individuals who have respiratory failure, septic shock, and/or multiple organ dysfunction.

**Table 2 ijerph-20-05182-t002:** Hazard rate and corresponding 95% confidence interval from a time-varying exposure (COVID-19) Cox PH model for repeated outcomes (injuries).

COVID-19	HR (95% CI)	*p*-Value
Yes vs. No	1.36 (1.05, 1.77)	0.020

**Table 3 ijerph-20-05182-t003:** Injury Incidence Rate, expressed as the expected number of injuries per 1000 h/player of exposure (training or competition). Estimates and comparisons were computed by a GLMM model.

COVID-19 Level	Incidence Rate (95% CI)	Comparisons	Relative Risk (95% CI)	*p*-Value
No COVID-19	2.87 (2.33; 3.52)	I/No COVID-19	0.94 (0.61; 1.46)	0.80
I	2.71 (1.77; 4.15)	(II–III)/No COVID-19	1.69 (1.21; 2.38)	0.002
II–III	4.86 (3.53; 6.70)			

**Table 4 ijerph-20-05182-t004:** Injury burden was defined as the total amount of days lost due to injuries for each player, scaled to 1000 h/player of exposure. Injury burden was estimated by modeling injury time for each patient using a GLMM with exposure time entered as an offset term.

COVID-19 Level	Injury Burden (95% CI)	Comparisons	Ratio (95% CI)	*p*-Value
No COVID-19	55.10 (39.78; 76.32)	I/No COVID-19	0.92 (0.54; 1.58)	0.77
I	50.75 (98.06; 88.64)	(II–III)/No COVID-19	1.86 (1.21; 2.86)	0.005
II–III	102.57 (66.48; 158.26)			

**Table 5 ijerph-20-05182-t005:** Estimates of the proportion of muscle injury, and corresponding CI 95%, according to different anatomical locations computed using a multinomial regression model.

COVID-19	Anatomical Location	Injury Probability (CI 95%)
No	Muscle–tendon junction	27.1% (21.3%; 32.9%)
	Muscle belly	72.9% (67.1%; 78.7%)
I	Muscle–tendon junction	35.3% (19.2%; 51.4%)
	Muscle belly	64.7% (48.6%; 80.8%)
II–III	Muscle–tendon junction	40.6% (28.6%; 52.7%)
	Muscle belly	59.4% (47.3%; 71.4%)

**Table 6 ijerph-20-05182-t006:** Average number of days of absence due to SARS-CoV-2 infection.

COVID-19 Level	Average Time (95% CI)	Difference (95% CI)	*p*-Value
I	16.18 (14.7; 17.6)	−0.02 (−1.16; 1.11)	0.97
II–III	16.16 (14.6; 17.7)		

## Data Availability

Data are available on reasonable request. Requests for data sharing from appropriate researchers and entities will be considered on a case-by-case basis. Interested parties should contact the corresponding author (dottor@alessandrocorsini.it).

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
