# Peer review of "Are Football Players More Prone to Muscle Injury after COVID-19 Infection? The “Italian Injury Study” during the Serie a Championship"

_ijerph, 2023, doi:10.3390/ijerph20065182_

Round 1

Reviewer 1 Report

Congratulations for your study.

The article is well-written and interesting. The authors described the muscle injury burden in the major Italian football league in the 2020-2021 season, during the second and third SARS-CoV-2 pandemic waves, describing any relationships with the infection.

Some suggestions and possible modifications can improve the paper:

- The abstract should be separated into sections following the journal's directions, making the aim clear from the first read.

- Some acronyms should be corrected (e.g. FIFA)

- Severe acute respiratory syndrome coronavirus 2 is the virus and it is abbreviated as SARS‑CoV‑2. Coronavirus disease 2019 is the relative respiratory illness and it is abbreviated as COVID-19. Please be consistent with names and capitalization throughout the paper.

- Did players with a moderate illness undergo to a maximal cardioulmonary exercise test for RTP? yes, it would be useful to report this data.

 -The authors rightly make no allusion to causal inferences concerning COVID-19 infection and injury rating but highlighted the correlation between COVID-19 infection and the increase in indirect muscle injuries. Moreover, no difference in time to RTP after infections were found with the athletes who suffered a COVID-19-related injury (level II and III) had a more severe injury burden compared to athletes who suffered a COVID-19-unrelated-injury. Therefore the authors could advance the hypothesis of greater caution for RTP after moderate and severe infections in professional soccer players.

Author Response

Dear reviewer 1

Thanks for your suggestion, we really appreciated your effort.

We modified the manuscript by adding what you suggested, thus improving the whole text

Reviewer 2 Report

This work investigates the likelihood of muscle injury in football players after Covid-19 infection during the Italian Serie A Championship. 433 football players were studied with 173 Sars-Cov-2 infections and 332 indirect muscle strains.

As mentioned, that “influence of COVID-19 level on the risk of injury” (line 81-82) has not been reported for football players in the literature, authors should have provided more references related to correlation between Covid infection severity and risk of musculoskeletal sequelae (for those studies not emphasis on football players). For example, Front Physiol., 2022, 13:813924, (https://doi.org/10.3389/fphys.2022.813924, noted that I am not in anyway related to those authors or the journal). This inclusion will give the general reader more information about the current state of the art in this area.

Based on my understanding, 433 subjects were studied. Out of those 433, 173 were infected with Sars-Cov2 with 104 had indirect muscular injuries. Conversely, within 260 subjects who were not infected, 228 had indirect muscular injuries. Please justify and comment the higher percentage of injuries for subjects who were not infected with Sars-Cov2 (228/260=87.7%) than those who were infected (104/173=60.1%). This observation seems to be contradicting to your conclusion.

Please also provide the actual number of injuries in each group (no Covid-19, level I, level II-III). Moreover, a more appropriate heading description for table 5 would be “proportion of injuries” rather than “injuries probability”, and again offering the number of injuries would be more meaningful than just percentage in table 5.

Suggest to revise the following sentence, “This study confirms the … muscle injuries.” (line 363-364). For example, the modified sentence could be, “Our results show the correlation between Covid-19 infection and the increase in indirect muscle injuries.” Please noted my stand is not against the hypothesis that Covid-19 infection increases the likelihood for indirect muscle injuries. However, agree with authors that more needs to be done to clarify the role of Covid-19 in causing any sequelae, especially football muscle injuries.       

Author Response

Dear Reviewer 2

Please find attached a detailed response to your comments

Thanks!

Round 2

Reviewer 2 Report

Thank all authors for looking into my comments and revise accordingly. This work could be published. Nevertheless, please explain and clarify in the script clearly similar to your reply "injuries can be a repeating event.... as soon as he develops the infections.". This clarification could avoid misconception for higher number of injuries for subjects who were not infected (paragraph 1 & 2 of results section, and table 5 in the revised manuscript).  

Author Response

Thanks. We addedd this information in our text

Thanks again for your time in the revision and implementation process of our study.

Best wishes